# Observation of triplet-assisted long-distance charge-transfer exciton transport in single organic cocrystal

Yejun Xiao [1] ✉, Xianchang Yan[1,2], Rui Cai[3], Xuan Liu[4], Jingwen Bao[1], Min Zhang[1,2], Jing Leng [1] ✉, Shengye Jin[1] ✉ & Wenming Tian [1] ✉

Charge-transfer (CT) states with long transport distances are highly desired for promoting the performance of organic optoelectronic devices in photoconversion and electroluminescence. However, due to the limited lifetime and small diffusivity, only nanoscale CT transport has been observed so far. Herein, taking a binary CT cocrystal (trans−1,2-diphenylethylene-1,2,4,5-tetracyanobenzene, named as $T_S$-$T_C$) with efficient thermally activated delayed fluorescence (TADF) as a model material, we report the direct observation of long-distance CT exciton transport by using modified time-resolved and photoluminescence-scanned imaging microscopy, which reveals a triplet-assisted CT transport mechanism. We demonstrate that, enabled by the long-lived and high-yield triplet state and efficient TADF, the average transport distance of over 80% of CT excitons in $T_S$-$T_C$ can be significantly enhanced from intrinsic nanoscale (≤58 nm) to ~11.2 μm. Our findings provide an effective strategy for greatly promoting short-lived CT exciton transport, which is of great significance for optoelectronic material design and device optimization.

Charge-transfer (CT) excitons, composed of bound electron-hole pairs delocalized across the adjacent donor-acceptor (D-A) interface, have been widely observed in organic semiconductor heterojunctions[1–6]. As important intermediates between tightly bound excitons and free carriers, CT excitons, especially their transport features, are central to functions like photoconversion and electroluminescence, making them crucial in various organic optoelectronic devices[7–12]. For example, in bulk-heterojunction organic photovoltaics (OPVs), the mobile CT states will facilitate the migration of CT excitons towards the lower-energy sites along the D-A interface, thereby providing additional opportunities for efficient exciton dissociation into free carriers[13,14]; while in third-generation organic light emitting diodes (OLEDs) with two host and one guest materials[1], the diffusive CT states may promote energy transfer to the diluted luminescent guest, thus improving the exciton harvesting and device performance.

With the rapid advancement of organic optoelectronic devices, the CT exciton transport, determined by their diffusivity and lifetime, has become the focus of both experimental and theoretical investigations[12,14–18]. However, due to the negligible absorption and weakly bound property of CT excitons, both the Förster and Dexter energy transfer (EnT) in D-A blends may exhibit relatively low efficiency, resulting in their small diffusivity[14–16]. On the other side, the presence of Coulomb interaction in CT excitons also leads to their finite lifetimes (often within nanosecond timescale). Therefore, the transport distances of CT excitons are usually confined within tens of nanometers as reported in various D-A blend systems[14,16,19]. In this

[1]State Key Laboratory of Chemical Reaction Dynamics, Dalian Institute of Chemical Physics, Chinese Academy of Sciences, Dalian, China. [2]University of Chinese Academy of Sciences, Beijing, China. [3]Instrumental Analysis Center, Dalian University of Technology, Dalian, China. [4]State Key Laboratory of Catalysis, Dalian National Laboratory for Clean Energy, Dalian Institute of Chemical Physics, Chinese Academy of Sciences, Dalian, China. ✉e-mail: yjxiao@dicp.ac.cn; ljx@dicp.ac.cn; sjin@dicp.ac.cn; tianwm@dicp.ac.cn

regard, developing an effective strategy for achieving long-distance CT exciton transport is highly desired for the potentially enhanced device performance.

A viable solution involves enhancing the carrier lifetime and/or diffusivity. The triplet excitons usually exhibit long lifetimes due to the spin-forbidden transition, enabling their long-distance diffusions over several micrometers[20–23]. Recently, Huang et al. reported a singlet-mediated triplet transport mechanism utilizing singlet fission, which significantly increases the triplet diffusivity by more than one order of magnitude[24,25]. Inspired by this cooperative singlet-triplet transport, we propose that, by employing the long-lived triplet state, the transport distance of CT excitons might also be significantly promoted. Due to the spatially separated highest occupied (HOMO) and lowest unoccupied molecular orbitals (LUMO), the singlet-triplet ($^1CT$-$^3CT$) exchange energy ($\Delta E_{ST}$) of CT excitons is notably narrowed, which facilitates the reverse intersystem crossing (RISC) process from $^3CT$ to $^1CT$ state and easily leads to the efficient emission of thermally activated delayed fluorescence (TADF)[1,26–28]. The efficient TADF of CT excitons, along with their long-lived $^3CT$ state, enables them to exhibit great potential for achieving long-distance CT exciton transport. To assess the potential of this triplet-assisted CT transport mechanism, a thorough understanding of the interplay between TADF and CT exciton transport is therefore in demand.

To address this issue, we endeavor to integrate the conventional phosphorescence decay recording method based on time-correlated single photon counting (TCSPC) technique with our PL-scanned imaging microscopy to realize the direct observation of CT transport on both nano- and milli-second timescales. On the other hand, CT cocrystals, composed of two or more donor and acceptor monomers, are primarily considered as promising candidates for exploring CT dynamics due to their well-defined structure, low defect density, and intense CT absorption[29–34]. In this work, taking a binary CT cocrystal (*trans*−1,2-diphenylethylene-1,2,4,5-tetracyanobenzene, named as $T_S$-$T_C$) with efficient TADF emission as a model material, we successfully observed a long-distance CT exciton transport and revealed a triplet-assisted CT transport mechanism. We demonstrated that, by utilizing a long-lived and high-yield $^3CT$ state, the average transport distance of over 80% of CT excitons in $T_S$-$T_C$ can be potentially increased by more than two orders of magnitude to ~11.2 μm.

## Results and discussion

### Optical absorption and emission of $T_S$-$T_C$

Ribbon-like $T_S$-$T_C$ cocrystal with a good phase purity was primarily synthesized following a previously reported solution drop-casting method (Supplementary Fig. 1, Fig. 2 and see Methods for more details)[35,36]. As revealed by the single-crystal X-ray diffraction (SXRD) analyses (details of the refinement data are provided in Supplementary Table 1), $T_S$-$T_C$ cocrystal exhibits the common mixed-stacking motif with a 1:2 D/A stoichiometry. Each benzene ring of TSB (*trans*-1,2-diphenylethylene) is overlapped by a TCNB (1,2,4,5-tetracyanobenzene) molecule with the measured D-A intermolecular distance of 3.38 Å (Supplementary Fig. 3). The short D-A distance can facilitate the π-electron delocalization from TSB to TCNB, leading to the intermolecular charge-transfer feature of $T_S$-$T_C$ (Fig. 1a). The absorption edge and the PL emission peak of $T_S$-$T_C$ are located at ~550 nm and ~570 nm respectively (Fig. 1b), both of which are apparently red-shifted compared with its individual components (Supplementary Fig. 4), demonstrating the notable CT feature of $T_S$-$T_C$ cocrystal.

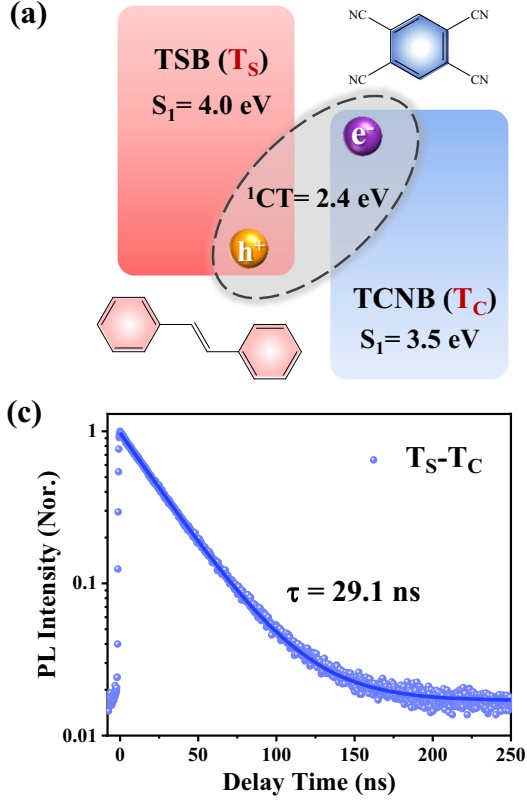

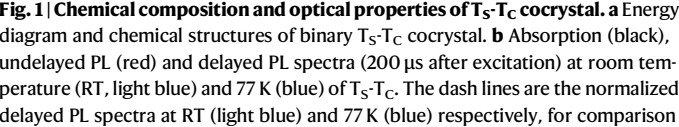

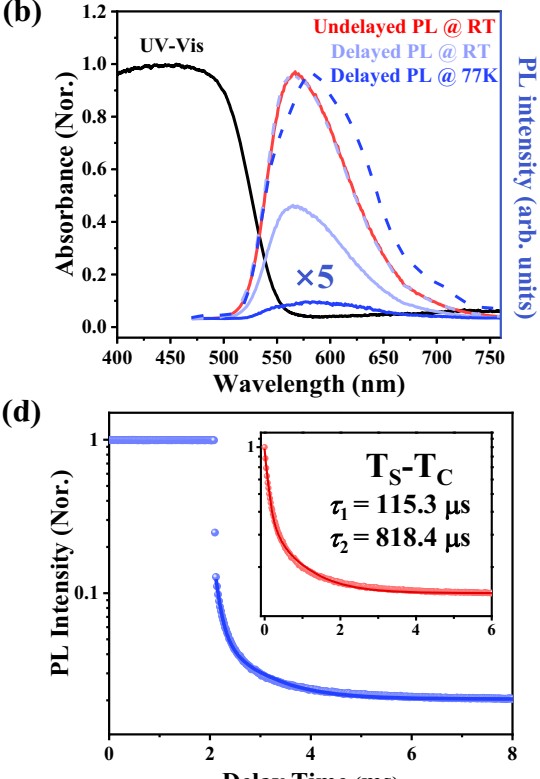

**Fig. 1 | Chemical composition and optical properties of $T_S$-$T_C$ cocrystal. a** Energy diagram and chemical structures of binary $T_S$-$T_C$ cocrystal. **b** Absorption (black), undelayed PL (red) and delayed PL spectra (200 μs after excitation) at room temperature (RT, light blue) and 77 K (blue) of $T_S$-$T_C$. The dash lines are the normalized delayed PL spectra at RT (light blue) and 77 K (blue) respectively, for comparison with the undelayed PL spectrum. PL kinetics collected on (**c**) nanosecond timescale and (**d**) millisecond timescale based on normal TCSPC technique and multipulse-excited TCSPC PDR technique, respectively. Insert of (**d**) is a delayed PL kinetics after deducting initial fluorescence interference. Solid lines in (**c**) and (**d**) are exponential fittings of these kinetics.

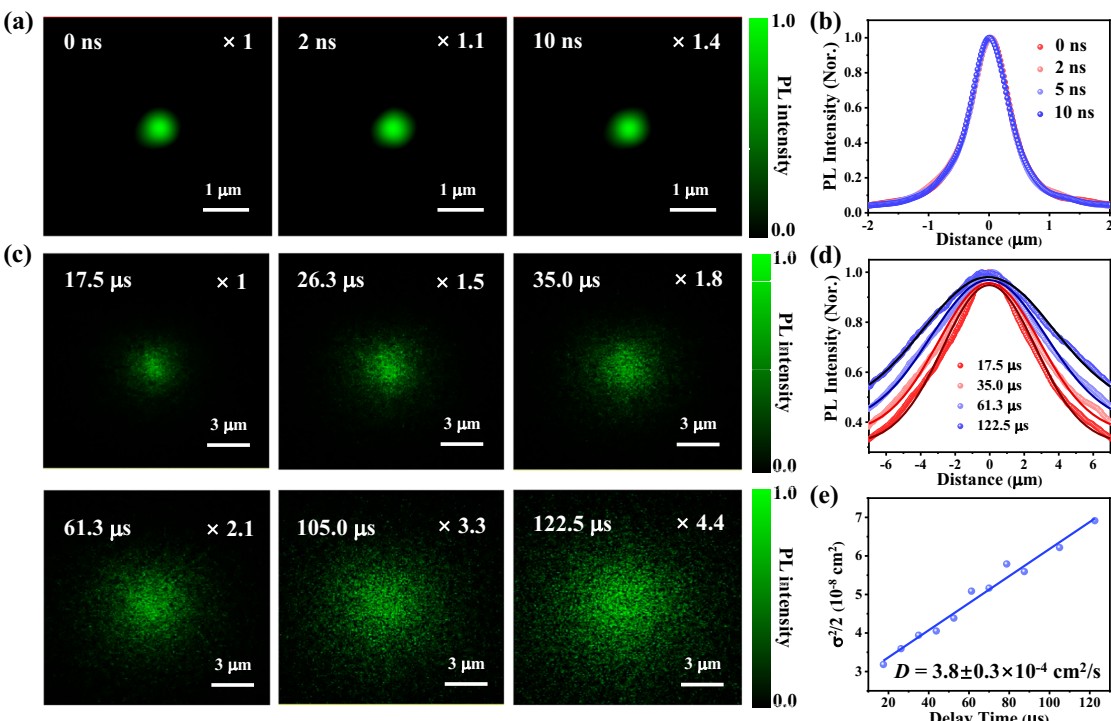

**Fig. 2 | Imaging and modeling of CT exciton transport.** PL intensity images from (**a**) fluorescence and (**c**) TADF emission in $T_S$-$T_C$ at different delay times after excitation. Scale bars are 1 μm and 3 μm, respectively. **b** Normalized 1D fluorescence intensity profiles of $T_S$-$T_C$ extracted from the x-axis in (**a**) at different delay times. The fluorescence distributions are almost identical, indicating the $^1$CT exciton transport distance in $T_S$-$T_C$ is within the spatial resolution (~58 nm) of ns-scale PL imaging measurements. **d** Normalized 1D TADF intensity profiles extracted from (**c**), along with their Gaussian fittings, showing the spatial broadening of the TADF-related exciton distribution. **e** The determination of TADF-related exciton diffusion coefficient by linearly fitting of the 1D time-dependent Gaussian variances ($\sigma^2(t)$).

To uncover the CT dynamics in $T_S$-$T_C$, we then carried out PL kinetic measurements by using TCSPC technique within both nano- and milli-second time windows (Fig. 1c, d). We note that, the ns-scale PL kinetics was collected by the typical TCSPC with a single-pulse excitation, while for PL kinetics on ms scale, a multipulse-excited TCSPC phosphorescence decay recording (PDR) technique was employed to improve the detection sensitivity (Supplementary Fig. 5; for more details see Methods)[37,38]. The fluorescence emission of $T_S$-$T_C$ exhibits a single exponential decay with a lifetime of ~29.1 ns (Fig. 1c). Besides that, a long-lived PL decay kinetics was also clearly observed on a ms timescale (Fig. 1d), which can be well fitted by a biexponential function, yielding two time constants of 115.3 μs and 818.4 μs (Supplementary Table 2). To avoid fluorescence interference during initial delay times, this data fitting started from the delay time of its second data point (17.5 μs, insert of Fig. 1d).

To clarify the origin of this long-lived emission, we further examined the delayed PL spectra (200 μs after excitation) at room temperature (RT) and 77 K, respectively. The delayed PL emission at RT exhibits an almost identical spectral profile to that of the total (undelayed) PL emission (Fig. 1b), and also similar decay kinetics collected at different wavelength ranges (Supplementary Fig. 6), indicating a typical TADF emission due to the RISC from triplet to singlet state. At 77 K, the delayed TADF emission was found to be prominently quenched and slightly red-shifted, suggesting a promoted phosphorescence emission at lower temperatures. Based on the relative shift between PL and phosphorescence emission peak (~570 nm and ~582 nm, respectively), the singlet-triplet energy gap ($\Delta E_{ST}$) is estimated to be ~45 meV, generally consistent with the previously reported value of ~10 meV[35]. Furthermore, to quantify the proportion of TADF in total PL emission at RT, we directly compared the integrated areas of total and delayed PL spectra (Supplementary Fig. 7a, 7b), from which the TADF weight can be estimated to be ~81%. This value is

consistent with the changes in total PL spectra area measured at RT and 77 K (~82%, Supplementary Fig. 7c), indicating an efficient TADF emission.

## Visualization and modeling of charge-transfer exciton transport

We next directly imaged the CT transport in $T_S$-$T_C$ by using time-resolved and PL-scanned imaging microscopy (see Supplementary Fig. 8 and Methods for more details)[39,40]. In brief, the excitation beam was focused to a near diffraction-limited spot (~550 nm in radius, Supplementary Fig. 9a) at a fixed position of $T_S$-$T_C$ through a 100× air objective (NA = 0.95), and the time-dependent PL intensity images can be obtained by scanning the PL collection spot on $T_S$-$T_C$ with a pair of galvano-mirrors. For ns-scale PL imaging, the local PL kinetics at any position can be extracted by the TCSPC module. While for PL imaging on the ms scale, the direct visualization of long-distance CT exciton migration was successfully realized using the aforementioned multipulse-excited TCSPC PDR technique.

Figure 2a shows the PL intensity images of $T_S$-$T_C$ collected at different delay times on a ns timescale. As the PL intensity is proportional to the exciton density, the time evolution of PL intensity distribution can directly reflect the exciton transport process, and the exciton transport in lateral dimension can be further described by a time-dependent two-dimensional (2D) Gaussian function[20,39]:

$$I_{PL}(x,y,t) \propto n(x,y,t) = N \exp\left[-\frac{(x-x_0)^2}{2\sigma_{x,t}^2} - \frac{(y-y_0)^2}{2\sigma_{y,t}^2}\right] \quad (1)$$

where $x_0$ and $y_0$ are the central position of Gaussian excitation beam, $\sigma_{x,t}^2$ and $\sigma_{y,t}^2$ are the time-dependent deviation of Gaussian profiles along x and y directions (see Supplementary Note 1 for detailed discussion). Exciton transport will result in the broadening of PL profile ($\sigma$) as delay time prolongs. By directly comparing their normalized

one-dimensional (1D) profiles of PL intensity (i.e., the cross section of PL image) at different delays, the exciton transport behavior can be more clearly observed and quantitatively determined. As shown in Fig. 2b, there is no obvious broadening of ns-scale PL (fluorescence) intensity profiles, indicating that the intrinsic $^1$CT transport distance in $T_S$-$T_C$ is within the spatial resolution of our measurements (~58 nm, Supplementary Note 1).

However, on a millisecond timescale, an apparent spatial expansion of CT exciton distribution can be clearly observed from the TADF intensity images with increased delay times (Fig. 2c), suggesting a notable exciton diffusion away from the initial excitation position. Specially, TADF intensity images after the second data point (t = 17.5 μs) were extracted and compared to circumvent the fluorescence emission during the initial delays. Due to a certain degree of exciton diffusion already occurred at t = 17.5 μs, the initial TADF distribution becomes much broader (~3.5 μm in radius) than the laser excitation spot (~550 nm in radius) (Supplementary Fig. 9). An isotopic exciton diffusion feature can be observed from the TADF imaging as shown in Fig. 2c, which may originate from the comparable distances between neighboring D-A units in $T_S$-$T_C$ across different directions (Supplementary Fig. 10). According to the solution to diffusion equation, the diffusion coefficient (D) of the isotropic CT transport can be directly obtained from a linear fitting of Gaussian variances of 1D PL profiles as a function of delay times (Supplementary Note 1)[24]:

$$D = \frac{\sigma_{x,t}^2 - \sigma_{x,0}^2}{2t} \tag{2}$$

The extracted 1D PL profiles of TADF intensity images can be well described by Gaussian functions (Fig. 2d), and the linear fitting of the time-dependent Gaussian variances yields $D = (3.8 \pm 0.3) \times 10^{-4}$ cm$^2$/s (Fig. 2e). It should be noted that the excitation intensity for TADF imaging was set to be low enough (i.e., 2.3 μJ/cm$^2$) to avoid the occurrence of higher-order recombination (Supplementary Fig. 11 and Fig. 12). In addition, we also performed a statistic analysis of the measured D values based on the TADF imaging of different $T_S$-$T_C$ cocrystals (Supplementary Fig. 13). The obtained diffusion coefficients are all in the range of 2.8 ~ 3.8 × 10$^{-4}$ cm$^2$/s, indicating the high crystal quality of $T_S$-$T_C$ cocrystals and the reliability of TADF imaging methodology employed in our measurements.

## Dynamic mechanism of charge-transfer excitons in $T_S$-$T_C$

Although the long-lived PL emission in $T_S$-$T_C$ has been well confirmed to originate from TADF, the related dynamic mechanism is still confusing considering the biexponential decay feature of TADF (Fig. 1d). Recently, Hu et al. found through theoretical calculations that the two lowest triplet states in $T_S$-$T_C$, $T_1$ (triplet state in TSB) and $^3$CT, are closely aligned in energy and thermally equilibrated with the $^1$CT state (Supplementary Fig. 14), and thus proposed a double-channel ISC mechanism for the biexponential TADF decay[35]. However, despite the assumed existence of two feasible ISC/RSIC pathways in $T_S$-$T_C$, our findings validate that the TADF kinetics should exhibit the monoexponential decay (see Supplementary Note 2 for detailed derivation), indicating the insufficiency of the double-channel ISC mechanism in explanation of the biexponential TADF decay in $T_S$-$T_C$[41].

Based on prior theoretical calculations[35], the relative energy levels and the possible carrier dynamic processes are shown in Fig. 3a. It can be seen that an energy order inversion occurs between $T_1$ and $^1$CT states after the photoexcited vibrational relaxation, which actually breaks the dynamic equilibrium between them. Therefore, TADF should be dominated by the RISC process from $^3$CT to $^1$CT, and the contribution of $T_1$-to-$^1$CT channel can be reasonably ignored. The $^3$CT dominated RISC can be verified by the large magneto-photoluminescence (MPL) effect (~8.6%) observed in $T_S$-$T_C$ (Supplementary Fig. 15), because Frenkel excitons with strong exchange

interaction are generally insensitive to the magnetic field (MPL ~ 1%)[42,43]. Subsequently, we carefully inferred the TADF kinetics and confirmed that the rate constant of TADF ($k_{DF}$) is determined by both the intrinsic $^3$CT lifetime ($k_3$) and the rate constants of ISC ($k_1$) and RISC ($k_2$) processes (Supplementary Note 2). To get further insights, delayed PL kinetics at different temperatures were compared in Fig. 3b. The decay profiles at different temperatures are almost identical with that of the phosphorescence emission from $^3$CT state at 77 K, suggesting a potential $k_3$-dominant TADF kinetics. Therefore, the CT transport observed via TADF imaging (Fig. 2c) is essentially a portrayal of the diffusion of $^3$CT excitons. The temperature-insensitive $k_3$ may be derived from the rigid crystal structure of $T_S$-$T_C$ as revealed by the comparison of crystallographic data measured at RT and 100 K, respectively (Supplementary Table 1)[44]. Moreover, it should be noted that although the $k_3$-dominaed TADF kinetics is temperature-insensitive, the TADF process in $T_S$-$T_C$ still exhibits temperature-dependence as evidenced by the monotonic decrease in PL intensity at lower temperatures (Fig. 3c). In addition, although the $^1$CT-$^3$CT conversion is normally forbidden in isolated D-A molecules, a $T_1$-mediated spin-vibronic coupling mechanism could probably enable this conversion in $T_S$-$T_C$ cocrystal[18,45].

The above discussion theoretically proposes that TADF should always exhibit a monoexponential decay feature. Therefore, we speculate that the measured biexponential TADF kinetics (Fig. 1d) likely originates from the diffusion of $^3$CT excitons[46]. For examination, we carried out the excitation-radius-dependent TADF kinetic measurements on $T_S$-$T_C$ (Fig. 3d). As expected, the fast component of TADF kinetics gradually disappeared as the excitation area changed from focused (initial TADF radius ~3.5 μm, Supplementary Fig. 9b) to defocused (initial TADF radius ~6.3 μm, Supplementary Fig. 16) and wide-field (full field of view, ~1000 × 800 μm$^2$) configuration (Fig. 3d), indicating a progressively eliminated diffusion effect due to the decreased exciton density gradient. According to a 2D exciton diffusion and recombination model (see Supplementary Note 3 for detailed discussion)[46], we further globally fitted the focused and defocused TADF kinetics, yielding the intrinsic $^3$CT lifetime of 850 μs and $^3$CT diffusion coefficient of 3.5 × 10$^{-4}$ cm$^2$/s (Fig. 3d). Based on the experimentally obtained parameters of $k_{PF}$ (3.44 × 10$^7$ s$^{-1}$), $k_{DF}$ (1.22 × 10$^3$ s$^{-1}$), and the prompt fluorescence and TADF proportions ($\varnothing_{PF}$ = 0.19 and $\varnothing_{DF}$ = 0.81, see Supplementary Fig. 7), the rate constants of $k_0$ ($^1$CT recombination), $k_1$, $k_2$ and $k_3$ can be further determined to be 1.0 × 10$^5$ s$^{-1}$, 3.43 × 10$^7$ s$^{-1}$, 4.88 × 10$^3$ s$^{-1}$ and 1.21 × 10$^3$ s$^{-1}$, respectively (Supplementary Note 2, illustrated in Fig. 3a). The experimentally determined intrinsic lifetime and diffusion coefficient of $^3$CT state (826.4 μs and 3.8 × 10$^{-4}$ cm$^2$/s shown in Figs. 3a and 2e, respectively) agree well with the theoretically fitted parameters of 850 μs and 3.5 × 10$^{-4}$ cm$^2$/s, respectively (Fig. 3d). These agreements not only confirm the fast component of TADF kinetics originating from the $^3$CT diffusion, but also indicate the high accuracy of TADF imaging measurements based on the multipulse-excited TCSPC PDR technique.

## $^3$CT-assisted long-distance $^1$CT exciton transport

Next, we aim to gain further insight into the CT transport mechanism. Three mechanisms may contribute to the observed spatial diffusion: photon recycling, energy transfer, and charge transfer. The re-absorption effect in photon recycling can be ruled out due to the low self-absorption coefficient of $T_S$-$T_C$ (Supplementary Fig. 17) and the invariant PL spectra profile at distances far from the excitation spot (Supplementary Fig. 18)[47–49]. Moreover, in typical D-A blends, both Förster and Dexter EnT of CT excitons were argued to be inefficient due to their negligible absorption and weakly bound property, and an asynchronous charge transfer mechanism known as the "inchworm" type transport has been proposed to elucidate the motion of CT excitons[14–16]. While in our $T_S$-$T_C$ model system, the stronger π-π interaction between D and A molecules (Fig. 1) may facilitate the direct

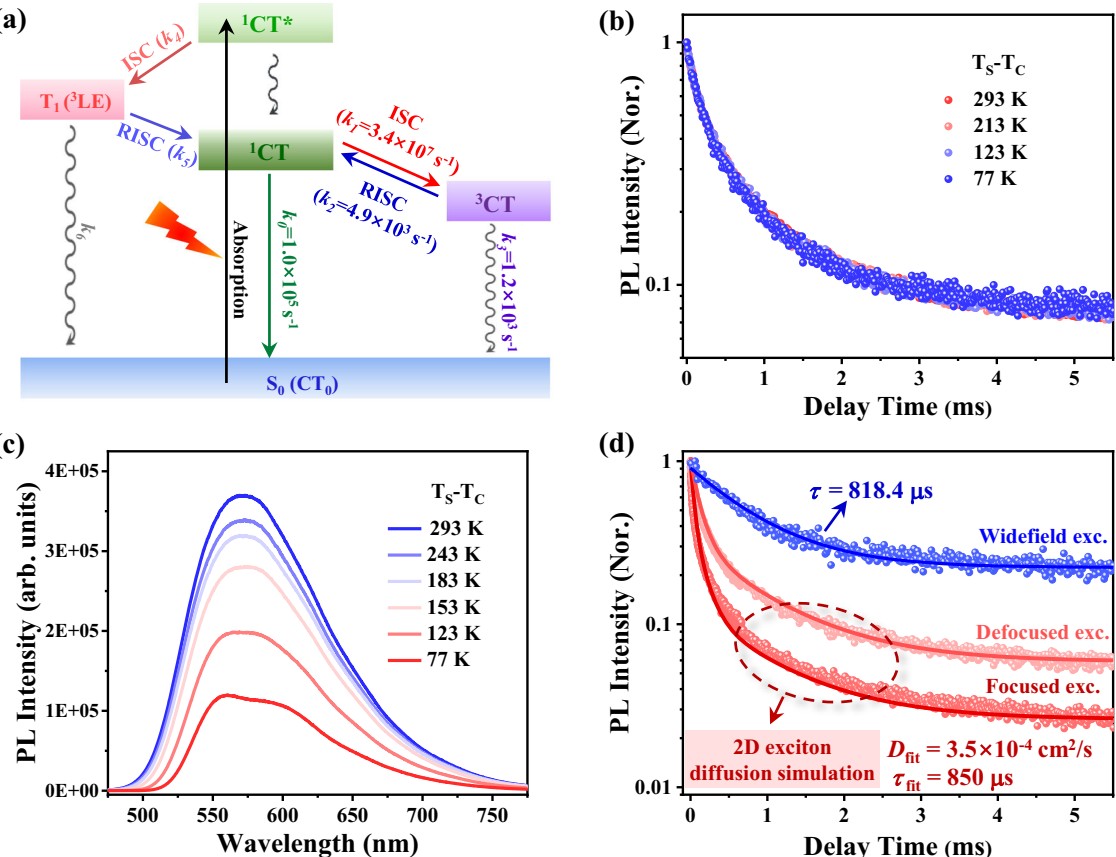

**Fig. 3 | Carrier dynamic processes in $T_S$-$T_C$ cocrystal. a** Schematic diagram for carrier dynamic processes in $T_S$-$T_C$ cocrystal. **b** Comparison of delayed PL kinetics collected at different temperatures under 375 nm excitation. These PL kinetics show almost no temperature dependence, indicating the sole TADF emission dominated by the $^3$CT decay kinetics. **c** Comparison of steady-state PL spectra collected at different temperatures under 375 nm excitation, indicating a typical temperature-dependent TADF emission feature. **d** Comparison of TADF kinetics in $T_S$-$T_C$ measured under focused (~3.5 μm in initial TADF radius), defocused (~6.3 μm in initial TADF radius) and widefield (in range of ~1000 × 800 μm²) excitation, showing a gradually eliminated fast decay component due to the reduced impact of $^3$CT diffusion. Blue solid line is the single-exponential fitting of the widefield kinetics, while the red and dark red ones are the global fits of focused and defocused kinetics in terms of 2D exciton diffusion model described in Supplementary Note 3. The excitation intensity was set to be 2.3 μJ/cm² for both focused and defocused excitation, and ~4.5 × 10⁻⁵ μJ/cm² for the widefield excitation.

CT transport in an excitonic form. The tightly bound nature of CT excitons in $T_S$-$T_C$ can be verified by an excitation-intensity dependent PL intensity measurement (Supplementary Fig. 19), and further corroborated by their exponential decay feature (Fig. 1a)[12]. Therefore, we are prone to retain the Förster and/or Dexter mechanism for demonstrating the CT transport in $T_S$-$T_C$.

The $^3$CT diffusion coefficient of $3.8 × 10^{-4}$ cm²/s aligns with the typical triplet diffusion coefficient range ($10^{-3}$–$10^{-4}$ cm²/s) observed in other organic semiconductors[21,24], indicating the Dexter EnT mechanism underlies $^3$CT transport in $T_S$-$T_C$. This speculation can be further verified by the PL imaging measurements performed at 243 K. The $^3$CT diffusion coefficient, determined to be ~$5.8 × 10^{-4}$ cm²/s at 243 K, is slightly larger than that of $3.8 × 10^{-4}$ cm²/s at RT, which confirms the temperature-dependent Dexter-type $^3$CT transport in $T_S$-$T_C$ (Supplementary Fig. 20). While for $^1$CT transport (Fig. 4a), the diffusion coefficient can be roughly estimated to be $≤3.1 × 10^{-4}$ cm²/s based on its lifetime (~29.1 ns) and diffusion distance ($L_D ≤ 58$ nm, $L_D = 2\sqrt{D\tau}$), indicating a potentially inefficient Förster EnT. Additional PL quenching measurements further yielded an estimated $^1$CT diffusion coefficient on the order of $10^{-4}$ cm²/s (see Supplementary Fig. 21 and Note 4 for detailed discussion)[50], which consists with the maximum $^1$CT $D$ value of $3.1 × 10^{-4}$ cm²/s, and is comparable to the magnitude of $^3$CT diffusion. Accordingly, we suppose that the $^1$CT transport in $T_S$-$T_C$ should also follow the Dexter EnT mechanism rather than the Förster EnT pathway.

In contrast, despite exhibiting a moderate diffusivity ($3.8 × 10^{-4}$ cm²/s), $^3$CT excitons in $T_S$-$T_C$ can realize a remarkable diffusion over ~11.2 μm because of their long lifetimes (826.4 μs). Meanwhile, due to the small $^1$CT-$^3$CT energy difference in $T_S$-$T_C$, the migrative $^3$CT excitons can regenerate into $^1$CT excitons through the effective RISC, thereby resulting in an equivalent long-distance $^1$CT exciton transport assisted by the $^3$CT state. Based on the above discussion, we proposed a $^3$CT-assisted long-distance CT exciton transport mechanism as illustrated in Fig. 4b. Despite the inefficient EnT and short lifetime of $^1$CT excitons limiting their intrinsic diffusion within 58 nm (Fig. 4a), the transport distance can be remarkably promoted beyond two orders of magnitude through the $^3$CT-assisted pathway, which is facilitated by the efficient TADF and long-lived $^3$CT state (Fig. 4b). Moreover, based on the previously estimated TADF proportion (Supplementary Fig. 7a), the triplet yield in $T_S$-$T_C$ was further inferred to be >81% (Supplementary Note 2), which indicates that over 80% of CT excitons in $T_S$-$T_C$ have an average diffusion distance of over 10 μm with the assistance of $^3$CT state.

The long-distance CT transport is highly desired for promoting device performance in photoconversion and electroluminescence, but hard to realize because of their small diffusivity and short lifetime in most D-A blend systems[12,14,19,51]. We believe that the proposed triplet-assisted CT transport mechanism here may provide an effective strategy to tackle this challenge. For CT excitons with effective TADF, their triplet states can serve as potential energy

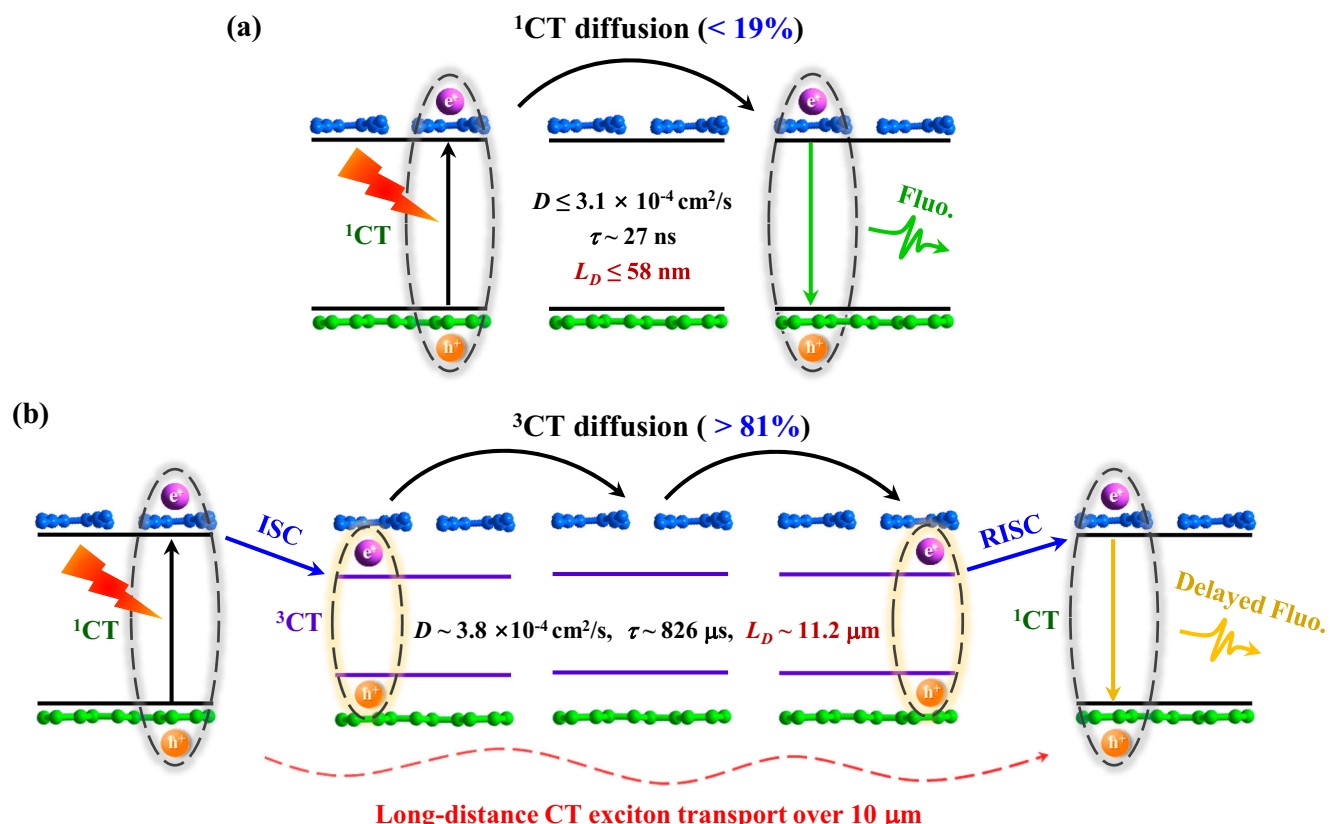

**Fig. 4 | CT exciton transport in T$_S$-T$_C$.** Schematics of (**a**) the intrinsic nanoscale $^1$CT exciton transport and (**b**) the triplet-assisted micron-scale CT exciton transport.

reservoirs where singlet excitons are quickly "stored" through the ISC process and undergo long-distance diffusion due to the long-lived feature of the triplet state, and then triplet excitons can return to the singlet via rapid RISC. Accordingly, a long-lived and high-yield triplet state combined with efficient TADF is integral to realizing the micron-scale CT transport. To verify the universality of the triplet-assisted CT transport mechanism, an additional TADF cocrystal (T$_X$-T$_C$, thioxanthone-1,2,4,5-tetracyanobenzene) was further synthesized[52,53]. Similar triplet-assisted CT transport over 5 μm was observed in this cocrystal (Supplementary Fig. 22), preliminarily confirming the universality of the proposed mechanism for enhancing the singlet diffusion in other TADF materials. The ability to achieve micron-scale CT exciton diffusion has important implications for optoelectronic applications. For instance, it holds great promise for resolving the longstanding contradiction between the nanoscale exciton diffusion lengths and the micron-scale optical absorption depths in OPVs[17,54,55]. Accordingly, detailed investigations into typical TADF film systems are currently in progress and will be presented in future work.

In summary, by coupling the multipulse-excited TCSPC PDR technique with PL-scanned imaging microscopy, we successfully visualized the long-distance CT transport in a binary T$_S$-T$_C$ cocrystal. The narrowed energy gap between $^1$CT and $^3$CT states not only facilitates the TADF emission of T$_S$-T$_C$, but also enables an efficient triplet-assisted transport channel. More than 80% of CT excitons can promisingly overcome the intrinsic nanoscale (≤58 nm) transport to achieve the long-distance migration over an average distance of 10 μm. We believe that this triplet-assisted, long-distance CT transport mechanism should also occur in other TADF materials with long-lived triplet states, which exhibits significant implications for improving exciton transport in TADF materials and their further applications in optoelectronics.

## Methods

### Synthesis of T$_S$-T$_C$ cocrystal

The T$_S$-T$_C$ cocrystals were synthesized by a solution self-assembly method. A mixture of TSB (18.0 mg, 0.1 mmol) and TCNB (35.6 mg, 0.2 mmol) was dissolved in 10 ml acetonitrile solvent and directly dropped of the mixed solution onto the substrate. Yellow ribbon-like T$_S$-T$_C$ cocrystals were obtained after the complete evaporation of solvent.

### Material characterizations

X-ray diffraction (XRD) pattern was obtained by using a X'pert Pro X-Ray Diffractometer (PANAlytical, Netherlands) using Cu Kα radiation. A scan rate of 5° min$^{-1}$ was applied in the range of 5–35°. Diffraction intensity data for single crystals of T$_S$-T$_C$ was collected on a GeminiUltra diffractometer equipped with graphite-monochromatic Mo Kα radiation ($\lambda$ = 0.71073 Å). The structure of T$_S$-T$_C$ was solved by direct methods and refined by full-matrix least-squares techniques based on F$^2$ using the SHELXS-97 programs[56]. All the non-hydrogen atoms were refined with anisotropic parameters, while hydrogen atoms were placed in calculated positions and refined using a riding model. UV–Vis diffuse reflectance spectra (DRS) were recorded using a UV–Vis spectrophotometer (JASCO V-550) equipped with an integrating sphere, and BaSO$_4$ powder was used as the reference for the baseline correction. Micro-UV–vis absorption spectra were recorded using a UV–Vis–NIR microspectrophotometer (CRAIC Technologies Inc., CRAIC 20/30PV Pro).

### PL and delayed PL measurements

The PL imaging and kinetic measurements on nanosecond time scale were performed on a home-built PL-scanned imaging microscopy coupled with a time-correlated single photon counting (TCSPC) module. The setup shown in Supplementary Fig. 8 contains both

wide-field (defocused) and focused illumination modules. A 375 nm pulse laser (PDL 800-B, PicoQuant) was focused on the sample through a 100× air objective lens (NA = 0.95, Olympus PLFLN) with the spot radius of ~550 nm. The excitation intensity is adjusted by a neutral density filter and measured with a power meter (PM100D S130VC, Thorlabs, USA). Fixing the excitation spot at a selected position on the sample, the PL emission from the whole sample can be collected by the fast rotation of a pair of galvanometer mirrors. Each scanning image contains 256 × 256 pixels (19 nm/pixel). The PL kinetics was collected by a high-speed detector (HPM-100-50, Hamamatsu, Japan) equipped with a 510 nm long pass filter. The steady-state PL emission spectra were obtained by a monochromator (SpectraPro-HRS-300, Princeton Instruments, USA) coupled with a charge coupled device (CCD) camera (PIXIS 100, Princeton Instruments, USA).

The delayed PL imaging and kinetic measurements were performed on the same setup with an on-off modulated 375 nm pulse laser for excitation. The repetition rate is 50 MHz, and the pixel time ($T_{pxl}$) is set to be 9 ms with laser-on time ($T_{on}$) of 3 ms and off time ($T_{off}$) of 6 ms. For wide-field (defocused) excitation, the excitation laser beam was uncollimated before the 100× air objective lens to form an excitation spot of ~10 µm in diameter. The scanning images of delayed PL contain 256 × 256 pixels (64 nm/pixel). The delayed PL kinetics was collected by a high-speed detector (HPM-100-50, Hamamatsu, Japan) equipped with appropriate band-pass filters. For measurements of temperature-dependent delayed PL kinetics, a 100× air objective lens with NA = 0.6 (Olympus SLMPlan N) was used instead. The delayed PL spectra were measured on a photoluminescence spectrometer (FLS1000, Edinburgh, UK). A 375 nm pulse laser was used for excitation, and the integration time window of the delayed PL spectra was set from 200 µs to 6 ms.

**Magneto-photoluminescence (MPL) measurements**

All MPL measurements were carried out in the vacuum chamber with a pressure <2.0 torr in a cryostat integrated with an electromagnet (OptiCool, Quantum Design). A 405 nm pulse laser (LDH-IB-405-B, PicoQuant) was used for excitation, and the spectra were acquired by a monochromator (SpectraPro-HRS-300, Princeton Instruments, USA) coupled with a charge coupled device (CCD) camera (PIXIS 100, Princeton Instruments, USA).

## Data availability

All raw data generated within the article and the Supplementary Information are available from the Source Data File. Source data are provided with this paper.

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

## Acknowledgements

We acknowledge support from the CAS projects for Young Scientists in Basic Research (YSBR-007 to S.J.), the Strategic Priority Research Program of the Chinese Academy of Sciences (XDB0970302 to W.T.), the National Natural Science Foundation of China (No. 22233005 to S.J., 22439001 to J.L., 22303105 to Y.X., 22122307 to W.T.), the Natural Science Foundation of Liaoning (2024JH3/50100010 to W.T.), the Liaoning Revitalization Talents Program (XLYC2203043 to W.T.), the Dalian Science and Technology Innovation Fund (2024RJ006 to W.T., 2024RQ021 to Y.X.) and the DICP funding (DICP I202206 to J.L., I202315 to W.T.). We thank Dr. Yuan CHENG from the Instrumentation and Service Center for Molecular Sciences at Westlake University for the assistance in micro-UV-vis absorption spectra measurements.

## Author contributions

W.T., S.J., J.L. and Y.X. conceived the project, Y.X. performed the sample preparation and characterization measurements, X.Y. designed the imaging technique, Y.X. performed the experiment and data analysis with the assistance X.Y., J.B. and M.Z., R.C. performed delayed PL measurement, X.L. contributed to the data analysis. The manuscript was drafted by Y.X. and revised by J.L., S.J. and W.T. All the authors discussed the results and contributed to the manuscript.

## Competing interests

The authors declare no competing interests.
