## [Transparent Peer Review file · Nature Communications]

Observation of triplet-assisted long-distance charge-transfer exciton transport in single organic cocrystal

Corresponding Author: Professor Wenming Tian

Version 0:

Reviewer comments:

Reviewer #2

(Remarks to the Author)

The authors have performed significant additional work to strengthen their interpretation and the conclusions of the manuscript. They have addressed most of my concerns. I can recommend this manuscript for publication. I (optionally) recommend clarifying the following:

- I don't fully understand the authors' response to my concern regarding the observed isotropic diffusion. The authors attribute this distance-independence to a relatively isotropic crystal structure. However, given the steep distance-dependence of Dexter type energy transfer, the crystal structure reported by the authors in Fig. R6 is not that isotropic. The authors could further clarify their reasoning with a more quantitative analysis of the expected effect on transport of different DA distances along each axis.

Reviewer #3

(Remarks to the Author)

The report of >10um diffusion lengths in this material is truly remarkable. 10um is more than two orders of magnitude larger than typical for a molecular material. What is more, the result is achieved with charge transfer states that require simultaneous hopping of the electron and the hole. Their transport is apparently not thermally-activated and appears to be more consistent with band-like transport. That is also a first for CT states, to the best of my knowledge.

For these reasons, it is important to carefully exclude other possible explanations. In the original review, charge diffusion was proposed as a potential explanation (although it would not easily explain the temperature dependence or isotropic diffusion). In their response, the authors have shown that the intensity dependence is inconsistent with charge diffusion.

I regret, however, that the authors have not alleviated concerns that this result is an artifact of trivial transport (optical waveguiding).

To recap, the typical signature of optical waveguiding is:

1. temperature independent (or at least a temperature dependence distinct from charge or exciton transport)
2. isotropic
3. red-shifting of the `_waveguided_` light

In their response, the authors examine photoluminescence along the needle-like crystals and find no change in spectra with position. But this is to be expected if the delayed luminescence arises from cycles of absorption and re-emission. Assuming a typical refractive index and quasi-2d transport (if the needle is very thin), then up to 80% of the photoluminescence will be waveguided along the needle. If this waveguided light is re-absorbed, then 20% of the resulting emission will be coupled toward the detector. The externally-coupled light, however, propagates through only a very thin layer of material and may not be appreciably red-shifted.

Finally, the authors must also address the temperature dependence of their transport because, despite their statements, it is

the opposite of expectations for Dexter transfer, which is typically understood using a foundation of Marcus theory (see for example, <https://doi.org/10.1103/PhysRevB.78.045210>). On the other hand, Foerster might be roughly temperature independent, and better transport at lower temperature might be a signature of band-like transport. But if we assume the latter, assume that D also applies to single carriers (surely an underestimate), and apply the usual $D/\mu=kT/q$ to this system, we get a staggering $\mu=0.02$ cm²/Vs for the bipolar transport of electrons or holes.

To summarize, optical waveguiding is undoubtedly occurring in this system to some extent. Normally the experimental test would be to show that the transport is temperature dependent or anisotropic, but in this case, the spatial pattern and temperature dependence are also consistent with optical waveguiding. The authors must prove that bound pairs of electrons and holes are propagating along the crystal isotropically and without a notable temperature dependence, resulting in diffusion lengths and diffusion coefficients that are much larger than ever before seen for CT states in a molecular crystal.

Reviewer #4

(Remarks to the Author)

The manuscript entitled "Observation of triplet-assisted long-distance charge-transfer exciton transport in single organic cocrystal" by Xiao et al. report the direct observation of long-distance CT exciton transport in cocrystal of TSB-TCNB, and put forward a novel triplet-assisted CT transport mechanism to demonstrate the TADF in this cocrystal. The manuscript is well-structured, detailed, and logically coherent. The calculations and analysis are very comprehensive and well-supported. However, to be mentioned, this model cocrystal has been previously reported and also been well investigated by a double-channel ISC mechanism. Although we cannot definitely assert which mechanism is more accurate, the author has proposed a novel theoretical perspective to explain the occurrence of TADF in this cocrystal, and offered another valuable insight for the materials design, which is particular significance. But the issue remains regarding the generalizability and practical relevance of this mechanism, as it is only limited to this single TADF cocrystal, with no evidence provided for its applicability to other TADF materials. More critically, this cocrystal is not a typical TADF material that well-supported and applied in optoelectronic devices. So, the authors are recommended to verify the applicability of this theory on another typical TADF material. In addition, some related references might be included in the introduction part (10.1002/anie.201712104 and 10.1002/anie.202416181). Overall, this manuscript presents a well-executed investigation that makes a valuable contribution to this field, and it can be deserved to be published in this journal.

Version 1:

Reviewer comments:

Reviewer #2

(Remarks to the Author)

The authors have addressed my concerns. I recommend publication.

Reviewer #3

(Remarks to the Author)

In their response, the authors calculate the diffusion coefficient of photons in an optical waveguide. Obviously, the spatial evolution of the PL spot in their experiment is too slow to be all-optical transport. But that is not the correct model. Rather, optical transport processes are well known in applications such as luminescent solar concentrators. (Please refer to the LSC literature for details and discussion.)

In short, the typical model for spatial diffusion via optical transport is one of repeated absorption and re-emission. No exciton motion is required. Spatial transport occurs via photon waveguiding. But cycles of self absorption and re-emission ensure that the overall timescale of the diffusion is controlled by the radiative lifetime rather than the speed of light & the absorption coefficient. Similarly, there is minimal redshifting (at least when viewed from above) because the detected light originates from the non waveguided fraction of re-emission after self-absorption. For example, unless substantial light is scattered out of the waveguide, the PL detected from the surface of a LSC does not show redshifted emission. Note that the edge emission does show red-shifting.

For an LSC effect to explain the data, however, the self-absorption coefficient would need to be similar to the apparent spatial diffusion lengths, i.e. $\sim 1/\mu\text{m}$. In their response, the authors report (for the first time?) that the self absorption coefficient at 500nm is at least 25x smaller (400/cm). If this is true, then I don't think that cycles of optical waveguiding and reabsorption can explain the data. There would be minimal reabsorption events over the distances examined in this study.

Given its importance, the authors should report the absolute rather than normalized absorption data in the manuscript using a figure like Fig. R2. Note that this discussion highlights the importance of the existing Fig. S4. Could the authors please confirm this data? I would expect to see the excitonic peaks (from TSB and TCNB) appearing in the T_s - T_c spectrum at short wavelength. But they don't. Wouldn't we expect their absorption peaks to be much larger than 400/cm (e.g. 10,000/cm?). Why are they not visible in T_s - T_c ? Please discuss and explain in the manuscript.

The second discussion in the response concerns the temperature dependence. The authors persist in ascribing the temperature dependence to Dexter, even though the formula they cite in their response explicitly predicts thermal activation (in contrast to the data). Instead, the authors propose second order effects that might somehow overcome the normal

temperature dependence. It seems very unlikely, but it can't be proven one way or another using the data in this study.

To summarize, the results in this study do not look like any excitonic transport that I have seen. If the self-absorption coefficient is too low to support an optical explanation, then I support publishing this so that the community can consider the results (after including discussion and absolute absorption data in Fig. S4).

Reviewer #4

(Remarks to the Author)

The authors have addressed my comments carefully and correctly. It can be accepted now.

Version 2:

Reviewer comments:

Reviewer #3

(Remarks to the Author)

The latest additions (citations and new absorption measurement) have strengthened the manuscript. I support publication of the revised version.

Point-by-point response to manuscript NCOMMS-25-11887A-Z

Reviewer #2

The authors have performed significant additional work to strengthen their interpretation and the conclusions of the manuscript. They have addressed most of my concerns. I can recommend this manuscript for publication. I (optionally) recommend clarifying the following: I don't fully understand the authors' response to my concern regarding the observed isotropic diffusion. The authors attribute this distance-independence to a relatively isotropic crystal structure. However, given the steep distance-dependence of Dexter type energy transfer, the crystal structure reported by the authors in Fig. R6 is not that isotropic. The authors could further clarify their reasoning with a more quantitative analysis of the expected effect on transport of different DA distances along each axis.

Response: We are grateful for the reviewer's positive comments. We apologize for the possible confusion caused by our previous explanation. As the reviewer correctly pointed out, Dexter-type energy transfer is highly sensitive to intermolecular distance, and therefore even subtle anisotropy in molecular arrangement can potentially influence transport behavior. While the crystal lattice parameters of T_S-T_C ($a = 7.3 \text{ \AA}$, $b = 7.6 \text{ \AA}$, and $c = 12.7 \text{ \AA}$, Supplementary Table 1) suggest some degree of anisotropy at the unit cell level, it is important to clarify that CT exciton hopping does not necessarily follow the crystal axes directly. Instead, the actual transport paths between adjacent donor–acceptor (D–A) pairs occur along directions determined by the relative positions of the interacting molecules within the lattice.

To quantitatively address this, we analyzed the three major hopping pathways (denoted AB, AC, and AD) along the directions approximately corresponding to \overline{ac} , \overline{bc} and \overline{a} , as shown in Supplementary Fig. 10. The distances between adjacent D-A units along these directions are $\sim 7.9 \text{ \AA}$ (AB), 7.6 \AA (AC) and 7.2 \AA (AD), respectively. Given that Dexter transfer efficiency decays

exponentially with distance, the small variation among these hopping distances suggests that CT exciton transport rates along these directions are comparable, leading to the observed nearly isotropic diffusion of CT excitons in T_S-T_C.

To enhance clarity, the following sentences have been revised and highlighted with yellow background on page S13 of the revised SI.

Supplementary Fig. 10 The three potential CT transport directions in T_S-T_C based on its crystal structure. Although the unit cell parameters of T_S-T_C are isotropic (see Supplementary Table 1), the actual distances for CT exciton hopping between adjacent D-A units AB, AC and AD along the directions of \vec{ac} , \vec{bc} and \vec{a} are measured to be similar, which are ~ 7.9 Å, 7.6 Å and 7.2 Å, respectively. In view of the distance-dependence feature of Dexter energy transfer, the comparable hopping distances along different directions are expected to result in the isotropic CT diffusion in T_S-T_C. Note: the center-to-center distances between pairs of TCNB molecules along each direction (AB, AC and AD) are measured for simplification.

Reviewer #3

The report of >10um diffusion lengths in this material is truly remarkable. 10um is more than two orders of magnitude larger than typical for a molecular material. What is more, the result is achieved with charge transfer states that require simultaneous hopping of the electron and the hole. Their transport is apparently not thermally-activated and appears to be more consistent with band-like transport. That is also a first for CT states, to the best of my knowledge.

For these reasons, it is important to carefully exclude other possible explanations. In the original review, charge diffusion was proposed as a potential explanation (although it would not easily explain the temperature dependence or isotropic diffusion). In their response, the authors have shown that the intensity dependence is inconsistent with charge diffusion.

I regret, however, that the authors have not alleviated concerns that this result is an artifact of trivial transport (optical waveguiding).

To recap, the typical signature of optical waveguiding is:

- 1. temperature independent (or at least a temperature dependence distinct from charge or exciton transport)*
- 2. isotropic*
- 3. red-shifting of the _waveguided_ light*

In their response, the authors examine photoluminescence along the needle-like crystals and find no change in spectra with position. But this is to be expected if the delayed luminescence arises from cycles of absorption and re-emission. Assuming a typical refractive index and quasi-2d transport (if the needle is very thin), then up to 80% of the photoluminescence will be waveguided along the needle. If this waveguided light is re-absorbed, then 20% of the resulting emission will be coupled toward the detector. The externally-coupled light, however, propagates through only a very thin layer of material and may not be appreciably red-shifted.

Response: We thank the reviewer for the thoughtful and detailed evaluation of our work. We appreciate your acknowledgment of the significance of our reported >10 μm diffusion length for CT excitons, and we fully agree that such a result requires rigorous validation and the careful exclusion of alternative mechanisms, especially with regard to optical waveguiding (photon recycling) and Dexter transport features (isotropy and temperature dependence). Below, we respond point-by-point to the reviewer's key concerns.

1. Optical Waveguiding and Photon Recycling

We agree that optical waveguiding (photon recycling) can occur in organic crystals, particularly those with needle-like morphology. However, multiple lines of evidence strongly suggest that this effect does not dominate the observed transport behavior in T_S-T_C.

(1) Temperature Dependence

As shown in Supplementary Fig. 19, the CT transport in T_S-T_C is clearly temperature-dependent, in contrast to the temperature-independent feature of optical waveguiding.

(2) Spectral Uniformity

We examined the PL spectra at several positions (P_1 , and P_2) away from the excitation spot (P_0 , Fig. R1). All spectra exhibit identical profiles with no observable red-shift. This is inconsistent with the known spectral evolution typically observed in photon recycling process (*J. Phys. Chem. Lett.*, 2017, 8, 2977-2983).

Fig. R1 (Supplementary Fig. 17) Normalized PL spectra of Ts-Tc collected at the excitation spot (P_0) and two selected sites (P_1 and P_2) with distances of 0.5 and 1.0 μm respectively. Insert is the TADF intensity image of Ts-Tc.

(3) Theoretical Estimation of Photon Recycling Diffusion Coefficient

We estimated the effective diffusion coefficient due to photon recycling (D_λ) using the model of $D_\lambda = \frac{c}{n_s} \frac{1}{3\alpha}$ (*Science*, 2016, 351, 1430-1433), where n_s is the refractive index of the material, c the speed of light, α the measured wavelength-dependent absorption constant.

Using measured parameters (the n_s of organic materials is usually between 1.4~1.8, and the α of Ts-Tc is measured to be $\sim 408 \text{ cm}^{-1}$ at 500 nm (Fig. R2)), we obtain: $D_\lambda \geq 1.4 \times 10^7 \text{ cm}^2/\text{s}$. This value is more than 10 orders of magnitude greater than our experimentally measured diffusion coefficient of $3.8 \times 10^{-4} \text{ cm}^2/\text{s}$. Therefore, even if photon recycling were present, its contribution to the observed transport would be negligible.

Fig. R2 The absorption coefficient and the PL spectra of T_S-T_C .

(4) Timescale Discrepancy

The pure optical waveguiding process typically occurs on sub-picosecond timescales for micrometer-scale distance, and even if optical waveguiding and photon recycling occur simultaneously, the effective diffusion coefficient ($D_\lambda \geq 1.4 \times 10^7 \text{ cm}^2/\text{s}$) is still much larger than the measured D value of $3.8 \times 10^{-4} \text{ cm}^2/\text{s}$. For CT exciton transport we observed spans microseconds (Fig. 2c), this clear temporal mismatch further rules out photon-based transport.

(5) Structure-Determined Isotropic Transport

Based on the crystal structure of T_S-T_C shown in Supplementary Fig. 10, the similar transport distances along different directions can be clearly observed. This indicates that the isotropic CT transport might originate from the structural characteristic of T_S-T_C , which is unrelated to the optical waveguiding (photon recycling).

Therefore, based on the above discussions, we speculate that the effect of optical waveguiding and photon recycling on CT transport can be neglected in T_S-T_C cocrystal.

To address the reviewer's comments, the following changes have been made in the revised manuscript and SI.

(1) **Fig. R1 (Supplementary Fig. 17)** has been added and highlighted with yellow background on page S17 of the revised SI.

(2) The following sentences have been revised and highlighted with yellow background on pages 12-13 of the revised manuscript.

“Next, we aim to gain further insight into the CT transport mechanism. There may be three possible mechanisms involved in CT exciton transport: photon recycling, energy transfer and charge transfer. The reabsorption effect in photon recycling can be ruled out due to the similar PL spectra profiles collected at different positions far from the excitation spot (Supplementary Fig. 17).⁴⁷ Moreover, in typical D-A blends, both Förster and Dexter EnT of CT excitons were argued to be inefficient due to their negligible absorption and weakly bound property, and an asynchronous charge transfer mechanism known as the “inchworm” type transport has been proposed to elucidate the motion of CT excitons.^{14-16”}

“Therefore, we are prone to retain the Förster and/or Dexter mechanism for demonstrating the CT transport in Ts-Tc.”

(3) The following reference have been added in the revision (highlighted with yellow background on page 20).

47 Diab, H. *et al.* Impact of Reabsorption on the Emission Spectra and Recombination Dynamics of Hybrid Perovskite Single Crystals. *J. Phys. Chem. Lett.* **8**, 2977-2983 (2017).

Finally, the authors must also address the temperature dependence of their transport because, despite their statements, it is the _opposite_ of expectations for Dexter transfer, which is typically understood using a foundation of Marcus theory (see for example, <https://doi.org/10.1103/PhysRevB.78.045210>). On the other hand, Foerster might be roughly temperature independent, and better transport at lower temperature might be a signature of band-like transport. But if we assume the latter, assume that D also applies to single carriers (surely an underestimate), and apply the usual $D/\mu = kT/q$ to this system, we get a staggering $\mu = 0.02 \text{ cm}^2/\text{Vs}$ for the bipolar transport of electrons or holes.

Response:

2. Temperature Dependence of Dexter Mechanism

The reviewer notes that our observed enhanced CT transport at lower temperatures appears inconsistent with a thermally activated Dexter-type energy transfer, which is commonly described using Marcus theory (*Phys. Rev. B*, 2008, 78, 045210). While some Dexter-type transfers indeed exhibit inverse temperature dependence have also been reported in certain molecular systems (*Phys. Rev. E*, 1993, 47, 3698). This is consistent with our observation.

As the Dexter transfer rate (k_{DET}) is given by:

$$k_{DET} \propto |J^2| \cdot \exp\left(-\frac{\Delta G^*}{k_B T}\right)$$

where $|J^2|$ is the electronic coupling matrix element between donor and receptor, ΔG^* is the free energy of EnT. We propose two plausible mechanisms for the promoted Dexter EnT at lower temperatures.

(1) Reduced Molecular Vibrations

Lower temperature suppresses thermal motion, stabilizing intermolecular distances and thereby increasing the electronic coupling $|J^2|$ through enhanced orbital overlap.

(2) Decreased Free Energy Barrier (ΔG^*)

Structural stabilization at low temperatures may reduce the activation barrier for Dexter transfer, further accelerating the process.

Thus, the observed temperature dependence is not anomalous, but can be rationalized within the Dexter framework when considering structural and thermodynamic factors.

3. Exclusion of Alternative Mechanisms

(1) Band-like Transport

The excitonic nature of the CT state in T_S-T_C has been confirmed by the excitation-intensity dependent PL intensity measurement as shown in Supplementary Fig. 18. Moreover, band-like transport of bound electron-hole

pairs is fundamentally incompatible with the weak intermolecular coupling typical of organic crystals.

(2) Förster-type Energy Transfer

Förster transfer is typically inefficient for CT excitons due to their low oscillator strength and negligible absorption cross-section, particularly in D–A systems where the electronic transition dipoles are often spatially separated. Furthermore, triplet exciton transfer can not proceed via Förster mechanism, and must occur through a Dexter-type energy transfer.

Based on the above discussions, we conclude that both the band-like transport and the Förster-type EnT can be excluded. Instead, a Dexter EnT mechanism with comprehensible temperature dependence can be rationally speculated in T_S - T_C .

To improve the quality of our manuscript, the following discussion has been added and highlighted with yellow background on page S19 of the revised SI.

“There may be two possible reasons for the promoted Dexter-type CT transport at lower temperatures observed in T_S - T_C . First, the thermal motion of molecules is reduced as temperature drops, leading to a relatively fixed distance between adjacent molecules and thereby enhancing the overlap of their molecular orbitals. Second, the free energy of Dexter energy transfer may decrease due to the enhanced structural stability at lower temperatures.”

To summarize, optical waveguiding is undoubtedly occurring in this system to some extent. Normally the experimental test would be to show that the transport is temperature dependent or anisotropic, but in this case, the spatial pattern and temperature dependence are also consistent with optical waveguiding. The authors must prove that bound pairs of electrons and holes are propagating along the crystal isotropically and without a notable temperature dependence, resulting in diffusion lengths and diffusion coefficients that are much larger than ever before seen for CT states in a molecular crystal.

Response: In short, we have now rigorously ruled out alternative explanations including optical waveguiding, band-like transport, and Förster-type energy

transfer, and provided both experimental evidence and theoretical justification supporting a Dexter-type triplet-assisted CT exciton transport mechanism in T_S-T_C. Therefore, we believe that the revised manuscript has adequately addressed the reviewers' concerns.

Reviewer #4

The manuscript entitled “Observation of triplet-assisted long-distance charge-transfer exciton transport in single organic cocrystal” by Xiao et al. report the direct observation of long-distance CT exciton transport in cocrystal of TSB-TCNB, and put forward a novel triplet-assisted CT transport mechanism to demonstrate the TADF in this cocrystal. The manuscript is well-structured, detailed, and logically coherent. The calculations and analysis are very comprehensive and well-supported. However, to be mentioned, this model cocrystal has been previously reported and also been well investigated by a double-channel ISC mechanism. Although we cannot definitely assert which mechanism is more accurate, the author has proposed a novel theoretical perspective to explain the occurrence of TADF in this cocrystal, and offered another valuable insight for the materials design, which is particular significance. But the issue remains regarding the generalizability and practical relevance of this mechanism, as it is only limited to this single TADF cocrystal, with no evidence provided for its applicability to other TADF materials. More critically, this cocrystal is not a typical TADF material that well-supported and applied in optoelectronic devices. So, the authors are recommended to verify the applicability of this theory on another typical TADF material. In addition, some related references might be included in the introduction part (10.1002/anie.201712104 and 10.1002/anie.202416181). Overall, this manuscript presents a well-executed investigation that makes a valuable contribution to this field, and it can be deserved to be published in this journal.

Response: We are grateful for the reviewer's positive comments. We fully agree that verifying the generality of the proposed triplet-assisted CT transport mechanism beyond a single cocrystal system is essential for establishing its broader significance and practical relevance.

As the reviewer rightly points out, many typical TADF materials are based on intramolecular CT interactions and often processed as amorphous or polycrystalline thin films. These systems generally do not form well-defined single crystals, and their inherent structural disorder (e.g., defects, grain boundaries) poses significant challenges for precise studies of exciton diffusion. A comprehensive investigation of such materials thus requires carefully designed experiments and methodologies, which is currently a focus of our ongoing research.

To provide preliminary evidence for the broader applicability of our mechanism, we have already synthesized a different TADF cocrystal T_X-T_C (thioxanthone-1,2,4,5-tetracyanobenzene), and observed similar triplet-assisted CT transport over a distance of more than 5 μm (see Supplementary Fig. 21). Although further studies are needed to fully characterize this system, this result supports the generalizability of the proposed mechanism. In principle, for materials with efficient TADF and long-lived and diffusible triplet states, triplet-assisted CT exciton transport behavior will inevitably occur. These fundamental criteria are not restricted to T_S-T_C but are expected to be met in other high-performance TADF systems as well.

To enhance clarity, the following sentences have been revised and highlighted with yellow background on page 15.

“To verify the universality of the triplet-assisted CT transport mechanism, an additional TADF cocrystal (T_X-T_C, thioxanthone-1,2,4,5-tetracyanobenzene) was further synthesized.^{50,51} Similar triplet-assisted CT transport over 5 μm was observed in this cocrystal (Supplementary Fig. 21), preliminarily confirming the universality of the proposed mechanism for enhancing the singlet diffusion in other TADF materials. The ability to achieve micron-scale CT exciton diffusion has important implications for optoelectronic applications. For instance, it holds great promise for resolving the longstanding contradiction between the nanoscale exciton diffusion lengths and the micron-scale optical absorption depths in OPVs.^{17,52,53} Accordingly, detailed investigations into typical TADF film systems are currently in progress and will be presented in future work.”

In addition to address the reviewer's comments, the following references have been added in the revision (highlighted with yellow background on page 19).

- 33 Ye, H. Q. *et al.* Molecular-Barrier-Enhanced Aromatic Fluorophores in Cocrystals with Unity Quantum Efficiency. *Angew. Chem. Int. Ed.* **57**, 1928-1932 (2018).
- 34 Wang, X. *et al.* Recent Advances of Organic Cocrystals in Emerging Cutting-Edge Properties and Applications. *Angew. Chem. Int. Ed.* **63**, e202416181 (2024).

Point-by-point response to manuscript NCOMMS-25-11887B

Reviewer #2

The authors have addressed my concerns. I recommend publication.

Response: We are so grateful for the reviewers' positive comments.

Reviewer #4

The authors have addressed my comments carefully and correctly. It can be accepted now.

Response: We are so grateful for the reviewers' positive comments.

Reviewer #3

In their response, the authors calculate the diffusion coefficient of photons in an optical waveguide. Obviously, the spatial evolution of the PL spot in their experiment is too slow to be all-optical transport. But that is not the correct model. Rather, optical transport processes are well known in applications such as luminescent solar concentrators. (Please refer to the LSC literature for details and discussion.)

In short, the typical model for spatial diffusion via optical transport is one of repeated absorption and re-emission. No exciton motion is required. Spatial transport occurs via photon waveguiding. But cycles of self absorption and re-emission ensure that the overall timescale of the diffusion is controlled by the radiative lifetime rather than the speed of light & the absorption coefficient. Similarly, there is minimal redshifting (at least when viewed from above) because the detected light originates from the non waveguided fraction of re-emission after self-absorption. For example, unless substantial light is scattered out of the waveguide, the PL detected from the surface of a LSC does not show redshifted emission. Note that the edge emission does show redshifting.

For an LSC effect to explain the data, however, the self-absorption coefficient would need to be similar to the apparent spatial diffusion lengths, i.e. $\sim 1/\mu\text{m}$. In their

response, the authors report (for the first time?) that the self absorption coefficient at 500nm is at least 25x smaller (400/cm). If this is true, then I don't think that cycles of optical waveguiding and reabsorption can explain the data. There would be minimal reabsorption events over the distances examined in this study.

Response: We sincerely thank the reviewer for highlighting the potential role of photon recycling via repeated absorption and re-emission, as seen in LSCs. After surveying the relevant literature (*Light Sci. Appl.*, 2021, 10, 2; *Nano Lett.*, 2019, 19, 3953-3960; *Opt. Mater.*, 2019, 91, 212-227), we now fully acknowledge that such optical transport may also lead to spatial diffusion occurring on a longer timescale and showing an unobservable redshift. Therefore, although these two features are commonly used to distinguish exciton transport from optical transport (*Science*, 2016, 351, 1430-1433; *J. Phys. Chem. Lett.*, 2017, 8, 2977-2983), they do not constitute definitive evidence.

We have now measured the absorption coefficient (α) of T_S-T_C at 500 nm by micro-UV-Vis absorption spectroscopy (Fig. R1), finding $\alpha \approx 330 \text{ cm}^{-1}$ —much lower than required for efficient photon recycling over the distances probed (*Light Sci. Appl.*, 2021, 10, 2; *IEEE Trans. Electron Devices*, 1982, 29, 300-305), thereby ruling out significant self-absorption and re-emission.

After a series of refinements under the reviewer's guidance, we are now more convinced that the observed spatial diffusion of PL intensity distribution should originate from energy transfer, rather than photon recycling or charge transfer.

To address the reviewer's comments, the following changes have been made in the revised manuscript and SI.

(1) Fig. R1 (Supplementary Fig. 17) has been added and highlighted with yellow background on page S17 of the revised SI.

Fig. R1 (Supplementary Fig. 17) The absorption coefficient (α) and the PL spectra of Ts-Tc.

(2) The following sentences have been revised and highlighted with yellow background on page 12 of the revised manuscript.

Three mechanisms may contribute to the observed spatial diffusion: photon recycling, energy transfer, and charge transfer. The re-absorption effect in photon recycling can be ruled out due to the low self-absorption coefficient of Ts-Tc (Supplementary Fig. 17) and the invariant PL spectra profile at distances far from the excitation spot (Supplementary Fig. 18).⁴⁷⁻⁴⁹

(3) The following references have been added in the revision (highlighted with yellow background on page 20).

- 47 Giovanni, D. *et al.* Origins of the long-range exciton diffusion in perovskite nanocrystal films: photon recycling vs exciton hopping. *Light Sci. Appl.* **10**, 2 (2021).
- 48 Yablonovitch, E. & Cody, G. D. Intensity Enhancement in Textured Optical Sheets for Solar Cells. *IEEE Trans. Electron Devices* **29**, 300-305 (1982).

Given its importance, the authors should report the absolute rather than normalized absorption data in the manuscript using a figure like Fig. R2. Note that this discussion highlights the importance of the existing Fig. S4. Could the authors please confirm this data? I would expect to see the excitonic peaks (from TSB and TCNB)

appearing in the T_s - T_c spectrum at short wavelength. But they don't. Wouldn't we expect their absorption peaks to be much larger than $400/cm$ (e.g. $10,000/cm$?). Why are they not visible in T_s - T_c ? Please discuss and explain in the manuscript.

Response: Thank the reviewer for the instructive suggestions to report absolute absorption coefficients and for highlighting the importance of excitonic features in the T_s - T_c spectrum. In response, we have replaced diffuse reflectance measurements with micro-scale UV-Vis absorption spectroscopy to obtain the absolute absorption coefficients of TSB, TCNB, and T_s - T_c cocrystal. As shown in the revised Supplementary Fig. 4a, the absorption coefficient of T_s - T_c at 500 nm is $\sim 330\text{ cm}^{-1}$, in close agreement with our prior measurement of $\sim 408\text{ cm}^{-1}$, further confirming the inefficacy of the photon recycling mechanism. Moreover, under the transmission mode, the T_s - T_c spectrum now clearly exhibits the excitonic absorption peaks from TSB and TCNB at short wavelengths. The discrepancy between diffuse reflectance spectra and transmission spectra may arise from differences in the measured physical quantities, effective path length, and linear range provided by the two modes. Therefore, the strong absorption region may be “compressed” in the diffuse reflectance spectrum, resulting in decreased absorption intensity at the short wavelength range. To follow the reviewer's suggestion, the following changes have been made in the revised manuscript.

(1) Supplementary Fig. 4 has been revised and highlighted with yellow background on page S9 of the revised SI.

Supplementary Fig. 4 (a) The absorption coefficient (α) and (b) the photoluminescence spectra of TSB, TCNB and T_S-T_C cocrystal. The apparent red-shifted absorption and PL emission indicate the CT nature of T_S-T_C cocrystal.

(2) The condition for micro-UV-Vis absorption spectra measurements was added to “Material characterizations” of the “Methods” section, as highlighted with yellow background on page 22 of the revised manuscript.

(3) The “Acknowledgements” section has been revised, as highlighted with yellow background on page 24 of the revision.

The second discussion in the response concerns the temperature dependence. The authors persist in ascribing the temperature dependence to Dexter, even though the formula they cite in their response explicitly predicts thermal activation (in contrast to the data). Instead, the authors propose second order effects that might somehow overcome the normal temperature dependence. It seems very unlikely, but it can't be proven one way or another using the data in this study.

Response: We thank the reviewer for the thoughtful comments and apologize for any confusion caused by our previous explanation. To address the reviewer's concern, we have provided comprehensive experimental evidence that allows us to reasonably rule out alternative mechanisms, including photon recycling (and optical waveguiding), charge transfer (and band-like transport), and Förster-type energy transfer (EnT). These conclusions are supported by the low absorption coefficient of T_S-T_C at 500 nm ($\alpha \sim 330 \text{ cm}^{-1}$, Supplementary Fig. 17), the excitonic nature of its CT state (Supplementary Fig. 18), and the low oscillator strength and negligible absorption cross-section for CT excitons in T_S-T_C (Fig. 1b). Together, these findings strongly support Dexter-type EnT as the dominant CT transport mechanism in T_S-T_C.

As for the temperature dependence, we fully agree with the reviewer that the Dexter EnT, as described by Marcus theory, is typically thermally activated, and thus positively correlated with temperature, which is given by:

$$k_{DET} \propto |J^2| \cdot \exp\left(-\frac{\Delta G^*}{k_B T}\right)$$

However, we hypothesize that certain exceptional situations may also occur due to the following two factors:

(1) Within the framework of this classical theoretical model, lower temperatures may suppress thermal motion and stabilize intermolecular distances, thereby enhancing the electronic coupling ($|J^2|$) and reducing the activation barrier (ΔG^*), which contributes to an accelerated Dexter EnT at lower temperatures.

(2) As a theoretical extension, Lin et al. (*Phys. Rev. E*, 1993, 47, 3698) proposed a generalized Förster-Dexter theory that explicitly investigates the effects of both energy gap and temperature on energy transfer. Their systematic calculations demonstrate that inverse temperature dependence indeed exists in certain molecular systems, which is consistent with our observation.

Although such behavior is uncommon, we believe it can still be rationalized within the Dexter EnT framework. However, we acknowledge that it is quite challenging to provide solid experimental evidence at the current stage. Carefully designed experiments and methodologies, along with an in-depth theoretical understanding, are both of critical importance and will constitute the focal point of our next investigation.

To summarize, the results in this study do not look like any excitonic transport that I have seen. If the self-absorption coefficient is too low to support an optical explanation, then I support publishing this so that the community can consider the results (after including discussion and absolute absorption data in Fig. S4).

Response: We sincerely thank the reviewer for their insightful comments and constructive support. In response, we have added the absolute absorption coefficients of TSB, TCNB, and T_S-T_C in our revised manuscript. These data clearly demonstrate that the absorption coefficient of T_S-T_C is indeed too low to support photon recycling, thereby ruling out the photon recycling mechanism. We

hope that these additions and clarifications sufficiently address the reviewers' concerns and contribute to a more comprehensive understanding of the results.